# Using Milk Thistle (*Silybum marianum*) Extract to Improve the Welfare, Growth Performance and Meat Quality of Broiler Chicken

**DOI:** 10.3390/ani12091085

**Published:** 2022-04-22

**Authors:** Wiktor Bendowski, Monika Michalczuk, Artur Jóźwik, Karwan Yaseen Kareem, Andrzej Łozicki, Jakub Karwacki, Damian Bień

**Affiliations:** 1Animal Science Students Scientific Association, Department of Animal Breeding, Institute of Animal Sciences, Warsaw University of Life Sciences, Ciszewskiego 8, 02-786 Warsaw, Poland; bendowskiwiktor@gmail.com (W.B.); jakubkarwacki18@gmail.com (J.K.); 2Department of Animal Breeding, Institute of Animal Sciences, Warsaw University of Life Sciences, Ciszewskiego 8, 02-786 Warsaw, Poland; karwan.kareem@su.edu.krd; 3Institute of Genetics and Animal Biotechnology PAS, Jastrzębiec, Postępu 36A, 05-552 Magdalenka, Poland; aa.jozwik@igbzpan.pl; 4Department of Animal Resource, Salahaddin University, Erbil 44002, Iraq; 5Division of Animal Nutrition, Institute of Animal Sciences, Warsaw University of Life Sciences, Ciszewskiego 8, 02-786 Warsaw, Poland; andrzej_lozicki@sggw.edu.pl

**Keywords:** broiler chickens, milk thistle, welfare, production results, quality, breast muscle, blood antioxidant

## Abstract

**Simple Summary:**

The latest trends in livestock husbandry and breeding are directed towards expanding production, which includes the use of natural products, such as herbs, in animal nutrition. The aim of the study was to determine the effects of an aqueous solution of milk thistle administered at chosen concentrations on the welfare, production results, selected physicochemical properties of the breast muscles, the activity of selected enzymes, and the antioxidant potential of the blood serum and breast muscles of broiler chickens. On the basis of the obtained results, it was found that both concentration levels of the milk thistle extract showed a significant effect on some of the growth performance, antioxidant capacity in the blood serum, and in pectoral muscle and welfare.

**Abstract:**

Modern poultry production looks for feed and water additives that improve the welfare of chickens. The utilization of herbs as additives in feed or drinking water is becoming increasingly well known. The aim of this study was to determine the effect of milk thistle extract (*Silybum marianum*), administered in drinking water, on chickens’ welfare, production results, selected physicochemical properties of the breast muscles, the activity of selected enzymes, and the antioxidant potential of blood serum and breast muscles. A total of 102 one-day-old male ROSS 308 chicks were allocated into three treatment groups. The material was an infusion of dried milk thistle seeds in two doses, administered in drinking water for chickens (group: C = 0; E1 = 0.24; E2 = 0.36 g/day/animal) under standard rearing conditions. As a result of adding the milk thistle to the drinking water, there was an improvement in the welfare of the chickens and in the production results, enzymatic capacity of selected enzymes, and antioxidant capacity in the blood serum and in pectoral muscle (*p* ≤ 0.05). The addition of a milk thistle seed infusion for chickens can be used in poultry production to improve the rearing performance antioxidant capacity and welfare of chickens.

## 1. Introduction

The utilization of herbs in animal production is in line with global trends for improving animal husbandry and breeding systems. The phytobiotic compounds present in herbs have a positive effect in improving health and increasing antioxidant potential, which contributes to the improvement of growth performance [1,2].

Milk thistle (*Silybum marianum* L.) is a cosmopolitan plant native to the Mediterranean. This plant is rich in bioactive substances such as silymarin and taxifolin [3]. Most of these compounds are found in the fruit and achenes [4]. Milk thistle seeds contain a number of substances that improve the homeostasis of the body, i.e., silibinin, silidianin, isosilichristin, silichristin, and isosilibin A and B [4,5]. Additionally, the high concentration of vitamin E in the seeds has a positive effect on antioxidant protection and metabolism [6]. Silymarin, along with other compounds contained in milk thistle seeds, have anti-hepatotoxic, antioxidant, anti-carcinogenic, and anti-inflammatory properties [7,8]. In addition, milk thistle, used in the diets of chickens, improves appetite, increases the secretion of digestive juices, and improves the functioning of the circulatory system and liver [9,10]. In previous studies, various authors have proposed different ways of administering milk thistle or its active ingredients to animals, in the form of seeds [11], cake [12], flour [13], or an extract [10].

In addition, it is worth taking note of the latest global trends that aim to reduce the number of medications used in animal production. The use of herbs in livestock therapy fits perfectly into the current direction of livestock production. The price of most easily available herbs for use in nutritional prophylaxis is definitely lower than most pharmacological substances. Taking into account the presence of the above-mentioned active compounds in milk thistle, it can be concluded that its use, in the right amount, will positively affect the health of chickens and the quality of the obtained product, meat. There is a paucity of information on the effects of milk thistle on the growth performance and welfare of broilers. Thus, the aim of this work was to examine the effects of milk thistle on growth performance, welfare, physicochemical properties, meat quality, and antioxidants in broilers.

## 2. Materials and Methods

### 2.1. Animals

In accordance with the Act of 15 January 2015, the 2nd Local Ethical Committee for Animal Experiments at the Warsaw University of Life Sciences (SGGW) decided that the described tests do not require the consent of the Ethics Committee (Justification 1 September 2021). The research material consisted of 102 male chicks of the ROSS 308 line, divided into three treatment groups. Each group had two replicates, with 17 birds per replicate; the treatment groups included: control (C), experimental 1 (E1), and experimental 2 (E2). The animals were fed with commercial, complete mixes: starter, grower, and finisher. Rearing lasted 42 days in standard conditions in chicken hose on floor system with wood pellets, with a light cycle were maintained according to ROSS 308 standard management guide [14] and access to fresh water. At the end of the experiment, the chickens were electrically stunned in a water bath (120 mA, 50 Hz) for 2 s. Next, they were slaughtered by the method of cutting the cervical blood vessels, and bled out for around 7–10 min. After scalding in water with a temperature of 56–58 °C for around 60 s, the birds were manually plucked and eviscerated, and their carcasses were placed in a cold store at a temperature of 4 °C for 24 h.

Water and feed were offered ad libitum to the birds until 42 d of age. During the experimental period, the broilers were fed according to the following feeding program: starter, days 1–10; grower I, days 11–24; grower II, days 25–35; and finisher, days 36–42 (Table 1).

### 2.2. Milk Thistle Dosage for Chickens

The factor that differentiated the experimental groups was the infusion of dried milk thistle, administered to the chickens in their drinking water from 7 to 42 days of age. The preparation of the infusion consisted of 4.08 and 6.12 g milk thistle seeds, which were added to 200 mL of water at 90 °C per each replicate for E1 and E2, respectively, and then applying the preparation to a bell drinker. Every day, that amount was prepared, which was started from the 7th day until the 42nd day of age; after finishing all waters, normal water was then offered again. The amount of raw material was determined on the basis of the lowest and highest daily intakes recommended by the manufacturer for an adult chicken and then converted to the standard weight of the animal at 42 days of age (Table 1).

### 2.3. Analyzed Parameters

#### 2.3.1. Growth Performance and Animal Welfare

Body weight was monitored at weekly intervals during the course of the experiment, and any decreases were recorded. On day 35, animal welfare was assessed using the gait score method, and 33 birds (*n* = 33) per excremental groups were taken randomly to gait score. During the evaluation, the chickens’ gait, time, and their crouching or lying down was analyzed. For the result to be objective, the assessment must be carried out by at least two people. The assessment was based on the proposed by Kestin et al. [15]:GS 0. No detectable abnormality, fluid locomotion, furled foot when raised;GS 1. Slight defect difficult to define;GS 2. Definite and identifiable defect, but it does not hinder the broiler in movement;GS 3. An obvious gait defect that affects the broiler’s ability to maneuver, accelerate, and gain speed;GS 4. A severe gait defect, the broiler will walk only a couple of steps if driven before sitting down;GS 5. Complete lameness, either cannot walk or cannot support weight on the legs.

Additionally, the foot pad was assessed for changes related to the occurrence of footpad dermatitis (FPD) [16], and 33 birds (*n* = 33) per excremental groups were taken randomly to gait score. Pododermatitis was assessed visually against a modified scale proposed by Rushen et al. [17]:FPD 0. No lesions;FPD 1. Superficial lesions, color lesions with a diameter not exceeding 0.5 cm;FPD 2. Deep lesions with a scab and ulceration, color lesions with diameter of 0.5 cm or greater.

Further, the feed conversion ratio (FCR) was calculated at the end of the rearing period.

#### 2.3.2. Assessment of Slaughter Efficiency, Chemical Composition, and the Physicochemical Properties of the Pectoral Muscles

Upon completion of the feeding experiment, 6 birds (*n* = 6) per experimental group were randomly selected and weighed prior to slaughter. After slaughter and cooling of the carcasses, the slaughter efficiency of the chickens was assessed on the basis of of the work of Michalczuk et al. [18], determining the percentages of pectoral muscle, legs, and giblets.

The basic chemical composition was determined for the collected samples of pectoral muscles: dry weight, crude fat, crude protein, and ash. The determinations were made using the NIR method [18]. The physicochemical properties of the pectoral muscles were analyzed (pH24 was determined with a pH-meter, CP-411, Elmetron, Zabrze). To determine the drip loss, the pectoral muscles were weighed; then, after 24 h, they were dried and reweighed. The drip loss was determined by the difference in mass. The water absorption coefficient was determined according to the Grau and Hamm method [19], while the meat color components were identified using a CR-410 colorimeter. The absolute color difference ΔE (between the breast muscle color of chickens in C and E1/E2) was calculated using the following equation:(1)$$\Delta E=\sqrt{{\left(L_{1}^{*}-L_{2}^{*}\right)}^{2}+{\left(a_{1}^{*}-a_{2}^{*}\right)}^{2}+{\left(b_{1}^{*}-b_{2}^{*}\right)}^{2}}$$
where
ΔE—absolute color difference;$L_{1}^{*}$, $a_{1}^{*}$, $b_{1}^{*}$—color parameters for the pectoral muscles of group C chickens;$L_{2}^{*}$, $a_{2}^{*}$, $b_{2}^{*}$—color parameters for the pectoral muscles of E1 or E2 chickens.

The obtained ΔE values were interpreted according to the scale:
0 < ΔE < 1—normal, invisible color deviation;1.01 < ΔE < 2—very small deviation, recognizable only by an experienced observer;2.01 < ΔE < 3.5—mean deviation, recognizable by an inexperienced observer;3.51 < ΔE < 5—clear color deviation;ΔE < 5.01—large color deviation.

#### 2.3.3. Indicators of Health Status and Antioxidant Potential

On the day of slaughter, six birds (*n* = 6) were randomly chosen to collect blood post-mortem in the amount of 1.5 mL per bird. Laboratory analysis aimed to also determine the activity of selected enzymes and antioxidant compounds by analyzing 5 g fragments of breast muscle taken from cooled carcasses.

In order to determine the effect of the infusion on the health of the chickens, the activity of the following enzymes from the blood and pectoral muscles of the chickens was determined: alanine aminopeptidase (AlaAP), leucine aminopeptidase (LeuAP), and arginine aminopeptidase (ArgAP), which are all responsible for limiting a harmful metabolism and accelerating protein circulation in the body, which translates into better weight gain. These amino acids were determined on the basis of McDonald and Barrett’s methodology [20].

The following compounds were also determined: the activity of acid phosphatase (AcP), beta-glucuronidase (BGRD), beta-galactosidase (BGAL), beta-glucosidase (BGLU), alpha-glucosidase (aGlu), mannosidase (MAN), and N-acetyl-BD-hexosaminidase (HEX). These compounds are responsible for breaking down complex sugars into simple sugars, as well as for the removal of harmful metabolites formed inside the cell. The measurements were made according to Barrett and Heath [21].

The influence of the infusion (in water) on the oxidative status of the collected breast muscles of broiler chickens was tested using the following determinations: vitamin C, glutathione (GSH), and 2,2-diphenyl-1-picrylhydrazyl (DPPH). Vitamin C concentration in serum and breast muscle was determined according to the method described by Michalczuk et al. [22]. The level of GSH was determined based on the immunochemical tests proposed by Tseng et al. [23]. For the quantitative colorimetric determination of GSH, a BIOXYTECH GSH-420 kit was used (OXIS Health Products, Inc. 6040 N CUTTER CIRCLE STE 317, Portland, OR, USA).

Measuring the radical scavenging activity was carried out using a routine test procedure employing the synthetic radical DPPH (1,1-diphenyl-2-picrylhydrazyl). Folin–Ciocalteu reagent was used as the oxidizing reagent and all chemicals were purchased from Sigma-Aldrich Chemie Gmbh (Munich, Germany) in the highest purity available.

#### 2.3.4. Statistical Analysis

The results were processed using the PS IMAGO PRO 5.1 statistical package and applying one-way ANOVA. The normality of the data was checked with the Shapiro–Wilk test. The homogeneity of variance was also checked with the help of Levene’s test for homogeneity of variance for ANOVA. Tukey’s post hoc test was used to determine the significance of the differences between the studied groups. When normality or homogeneity of variance tests failed, the nonparametric Kruskal–Wallis test was used. In case of group effect, pairwise comparisons were carried out using the Kruskal–Wallis test, with the Bonferroni method. The correlation coefficient between the selected variables being studied was calculated using the Kendall tau-b.

## 3. Results

### 3.1. Wellfare

The gait analysis of the chickens (GS) showed that the best gait index was obtained for birds given the milk thistle infusion (Table 2). The Kruskall–Wallis test found that both experimental groups E1 and E2 differed significantly from group C (C and E1, *p* = 0.014; C and E2, *p* = 0.007). In the E1 group, nearly 48.48% of the studied birds obtained a score of 0, and in the E2 group, it was almost 45.45%, while in the control group, 15.15% of chickens had 0. In the case of the visual evaluation of the chickens’ foot pads (FPD), significant differences were found for the C and E2 groups (*p* = 0.080), and the E1 and E2 groups (*p* = 0.080). Milk thistle administered in drinking water in the amount of 0.36 g/day/animal significantly reduced the number of feet scoring 1 or 2, i.e., feet where changes in the foot pad were visually observed.

The Kendall correlation coefficient is significant (FPD: *p* = 0.030; GS: *p* = 0.040), low, and negative (tau-b GS = −0.18; FPD = −0.27), which means that with an increase in the concentration of milk thistle in the infusion administered in the drinking water (g/day/animal), the assessment value obtained during the GS and FPD testing (for welfare level) more often decreased.

### 3.2. Production Results, Carcass Yield, Chemical Composition, and Quality of Meat

Significant differences were found in the final body weights (*p* = 0.004) and body weight gain (*p* = 0.033), with the E1 group achieving the highest body weight. Reasonably significant differences also occurred in the weight of the carcasses (*p* = 0.045). The carcasses from the E1 and E2 groups had a greater weight than those from group C. Among the slaughter yield parameters, significant differences were found in the slaughter yield, which was higher in C (*p* = 0.010) compared to the E1 group (Table 3). Significant differences were found in the proportion of abdominal fat, the highest share of which was in group C and the lowest in E2 (*p* = 0.039). No significant differences were found in the case of other parameters.

In the analyzed physicochemical parameters of the pectoral muscles (Table 4), significant differences were only found in the case of pH24, which was significantly higher in the E1 group than in the C group (*p* = 0.024). Higher color values were only observed in the E1 and E2 groups, but these differences were not statistically confirmed (*p* > 0.05). The ΔE parameter’s values prove the color stability of the pectoral muscles of the birds that received the addition of milk thistle infusion for drinking. The same applies to the comparison between the experimental groups E1 and E2. Under conditions of commercial breeding, adding the milk thistle infusion should not change the color of the meat from the consumer’s perspective.

### 3.3. Indicators of Health Status and Antioxidant Potential

Enzyme activity and indicators for antioxidant potential of the breast muscles are presented in Table 5. Significant differences were found for the E2 group compared to the control group, with statistically higher results observed for ArgAP (*p* = 0.044) and BGAL (*p* = 0.025). The E1 group showed significantly higher results than group C for BGLU and HEX (*p* = 0.006). The E1 group was characterized by a significantly higher MAN content (*p* = 0.033) than both groups C and E2. Group C had the highest content of ascorbic acid. A significant difference in this value was found between groups C and E2 (*p* = 0.045).

Enzyme activity and indicators for antioxidant potential of blood serum are presented in Table 6. Broiler chickens from the E1 group showed significantly higher enzymatic activity compared to the control group for AlaAP (*p* = 0.006), LeuAP (*p* = 0.031), ArgAP (0.018), and for ascorbic acid (*p* ≤ 0.018). Compared to the E2 group, the E1 group demonstrated significantly higher results for LeuAP (*p* = 0.031) and ArgAP (*p* = 0.018). The E2 group achieved statistically significantly higher results compared to the E1 group for BGAL (*p* = 0.025), BGLU (*p* = 0.018), and HEX (*p* = 0.014). However, compared to group C, the E2 group obtained statistically higher results for aGlu (*p* = 0.035) and GSH (*p* = 0.040). Both group C and E2 obtained statistically significantly higher DPPH results than the experimental group E1 (*p* = 0.011).

## 4. Discussion

Apart from the visual assessment of foot pad quality in terms of FPD changes, one of the methods to assess the welfare of chickens is the gait score test [15]. The treatment birds participating in this study obtained best gait ratings, enabling them to show natural behavior. Significant differences were found between groups C and E1 as well as C and E2 (*p* ≤ 0.05). The best gait index indicating for the birds administered with milk thistle may be because of the adding milk thistle and increasing dose of milk thistle from 0.24 g to 0.36 g that will reduce the severity of FPD. The gait results obtained for the control group may be because these chickens had the lowest body weights [24,25], as the vast majority of these birds (61%), despite not having access to the milk thistle infusion, scored 0 or 1. Improved gait quality associated with lower body weight has also been confirmed by other authors [26,27].

Inflammation of the foot pad (FPD) significantly affects the welfare of chickens, production results, and the selling price of livestock; hen pododermatitis is a very desirable raw material in Asian countries. The genesis of FPD is complex, as it is believed that it may be due to several factors, but it is related particularly to litter moisture, lack of balanced nutrition, and genetic susceptibility. A conceivable cross-link between the dietary treatments and litter quality may well be inferred due to the changes observed in bacterial composition of cecal digesta, which to an incredible extent contributed to the general litter conditions. The disease is caused by inflammation that affects the epidermis and the superficial layer of the skin. The disease begins with micro-trauma followed by a bacterial infection. Insufficient blood supply to the bird’s pododermatitis causes necrosis and keratosis of the epidermis, especially in the last period of the bird’s growth. Therefore, in this study, both GS and FPD assessments were carried out in the last stage of rearing, when birds are most susceptible to motion disturbances due to high body weight and litter parameters [16]. It was found that birds that received a drinking infusion of 0.36 g/day/animal differed significantly (*p* < 0.05) from group C in the foot pad quality assessment. In addition, the obtained results for all groups, in which the majority of chickens scored 0, testify to the proper welfare and appropriate microclimate conditions present during the rearing period. The maximum concentration of milk thistle infusion, administered in drinking water, may reduce FPD lesions.

It was found that milk thistle infusion significantly increased the weight of the chickens in groups E1 and E2. Birds fed milk thistle by 0.24 dosage had the highest body weight (*p* < 0.01) which is significantly more than birds fed with E2 and control group. Moreover, body weight before the 24th day of age was not significantly different (*p* > 0.05) between treatments. Conversely, after 24 days of age, the body weights for birds fed with milk thistle were significantly (*p* < 0.01) higher in comparison to the control.

Additionally, it was found that the addition of milk thistle in the E1 group reduced the feed conversion rate by 1.9% compared to the control group. However, the addition of an increased dose of milk thistle in the E2 group increased this coefficient; in the case of this experiment, the coefficient increased by 6.37% compared to the control group. These results may have been due to the presence of active compounds in the milk thistle seeds, mainly silymarin, which has a positive effect on liver prophylaxis antioxidant, as well as anti-carcinogenic properties [7,8].

Chickens receiving the milk thistle infusion in their drinking water had a 1 or 2 percentage points lower slaughter efficiency (depending on the group) compared to the control group. Similar results have been described in works by other authors investigating the effects of silymarin on slaughter yield [28]. The carcass weights of treatment groups E1 and E2 were significantly higher in comparison with control group. Moreover, the weight of breast and leg muscles in E1 and E2 were higher than C. There were no significant differences between E2 and C. Conversely, slaughter yield for control group was higher than E1. This finding is consistent with the work of Šťastník et al. [29], who reported that the highest slaughter yield was shown with the control group. The conducted research showed a lower proportion of abdominal fat in groups E1 and E2. It is estimated that the content of the carbohydrate fat accounted for 20% of the total body fat mass of a chicken [30,31]. Chicken broilers tend to store this fat during growth, but this type of fat is considered undesirable [32] and can cause economic losses for the producer [33]. Thanks to the addition of milk thistle extract to the water, the proportion of this fat was decreased. This may have been due to better metabolism and improved liver function [34]. It has been found that the addition of milk thistle expeller reduced blood cholesterol levels, which, according to other authors, improves the health of the birds [35].

Differences between the groups regarding pH24 were found when analyzing the quality parameters of the meat. The addition of milk thistle infusion reduced the amount of hydrogen ions in the analyzed pectoral muscles, which resulted in better quality meat. Meat acidity determines the technological usefulness of the meat—it determines the ability of the meat to retain its own water, tenderness, and color. If pH24 is below 5.72, chicken meat has a lower ability to bind water and retain it during thermal processing [36]. The E1 group was the only one to have the desired range of hydrogen ion content.

The metabolic processes in the cells of the body of chickens depend on the rate of synthesis and degradation of basic energy compounds: proteins, fats, and carbohydrates [37]. These processes are controlled by, inter alia, the activity of glycosidase enzymes [38]. The increased number of aminopeptidases due to the addition of milk thistle infusion increased the circulation of proteins in the chickens’ bodies. Increased protein circulation results in the improved growth performance of the animals, which can be seen in groups E1 and E2. In addition, the higher aminopeptidase content found in the breast muscles of the birds from the E1 and E2 groups may indicate that milk thistle has a positive effect on the maintenance of the homeostasis of the breast muscles. [39,40]. Beta glucosidase is involved in the removal of non-reducing glycosyl residues from saccharides and glycosides [41]. They are also involved in the metabolism of glycolipids [41]. The addition of milk thistle caused an increase in BGRD in the blood serum, which had a positive effect on the functioning of the chicken organism due to the decomposition of potentially harmful compounds.

Meat that is rich in polyunsaturated fatty acids is very sensitive to lipid oxidation [42]. Some of the oxidized lipid products may adversely affect the health of chickens and can display mutagenic, cytotoxic, or carcinogenic properties [43]. The effects of milk thistle infusion on groups E1 and E2 reduced the risk of undesirable oxidation. The lack of changes in DPPH in the breast muscles indicated the maintenance of a similar level of oxidative stress for all groups, which translated into being able to obtain a greater amount of valuable raw material from groups E1 and E2. In addition, there was a decrease in DPPH in the blood serum of 5.58 percentage points for the E1 group and 0.38 percentage points for the E2 group. Milk thistle meal used by other authors also positively influenced the tested parameter [13].

An increase in the glutathione levels [44] and vitamin C in the blood serum of the E1 and E2 groups testifies to the better oxidative stability of the organism. It has been shown that both of these antioxidants can synergistically interact [18]. In addition, a greater proportion of ascorbic acid in the blood serum caused by the addition of milk thistle has an anti-stress effect that directly affects the welfare of chickens and the quality of the raw material obtained [45].

## 5. Conclusions

The data obtained during the experiment show that the addition of milk thistle extract to drinking water for broiler chickens has a positive effect on the performance of the animals. In the case of both experimental groups, a higher level of welfare, greater production results, improved physicochemical properties of the breast muscles and the higher activity of selected enzymes, and the antioxidant potential of blood serum and breast muscles were found in comparison to chickens from the control group. The addition of milk thistle reduced FPD incidence for the experimental groups and was significantly lower than the control group. The addition of milk thistle seed infusion for chickens can be used in poultry production to improve the rearing performance and welfare of chickens.

## Figures and Tables

**Table 1 animals-12-01085-t001:** Formulation and nutritional composition of the chickens’ diets.

Item	Starter	Grower I	Grower II	Finisher
Ingredient (%):				
Wheat	35	35.03	35.6	30.12
Maize	24.2	30.00	30.42	38.00
Soybean meal	32.74	28.6	25	22
Rapeseed meal	3.0	0.0	0.0	2.0
Sunflower oil	1.3	1.9	3.0	3.2
Sunflower meal	0.0	2.0	3.0	3.0
Limestone	1.36	0.91	0.7	0.57
Monocalcium phosphate	0.7	0.38	0.12	0.06
Methionine	0.34	0.28	0.26	0.2
Sulfate of lysine	0.4	0.42	0.42	0.4
NaCl	0.2	0.2	0.2	0.2
Sodium sulfite	0.2	0.2	0.2	0.2
Threonine	0.06	0.08	0.08	0.05
Vitamin–mineral premix ^1^	0.5	0.5	0.5	0.5
Calculated chemical composition:				
AEM * (kcal/kg)	2901.2	3014.5	3108.5	3160.8
Crude protein %	22.36	20.51	19.45	18.43
Crude fat %	3.25	3.94	5.02	5.50

^1^ 1 kg of vitamin mineral premix contains: vitamin A, 10,000,000 IU; vitamin D3, 70,000 IU; HyD, 7500 mcg/kg; vitamin E, 25,000 mg/kg; vitamin K3, 600 mg/kg; thiamine B1, 600 mg/kg; riboflavin B2, 1600 mg/kg; pyridoxine B6, 800 mg/kg; niacin, 10,000 mg/kg; calcium pantothenate, 3266 mg/kg; biotin, 40,000 mcg/kg; choline chloride, 69,565 mg/kg; zinc, 20,000 mg/kg; iron, 10,000 mg/kg; copper, 4000 mg/kg; iodine, 200 mg/kg; selenium, 200 mg/kg; manganese, 20,000 mg/kg. * AEM = apparent metabolizable energy.

**Table 2 animals-12-01085-t002:** Distribution of assessed animal-based welfare indicators (in percent) in the 3 groups, including significant differences (*n* = 99).

	Statistic	Score	Group					
			C		E1		E2	
	Indicator		%	*n*	%	*n*	%	*n*
Footpad dermatitis		0	54.55	18	69.70	23	84.85	28
		1	36.36	12	30.30	10	15.15	5
		2	09.09	3	0.00	0	0.00	0
Kruskal–Wallis test	*p* = 0.010		A		AB		C	
Gait score		0	15.15	5	48.48	16	45.45	15
		1	45.45	15	24.24	8	36.36	12
		2	33.33	11	27.27	9	18.18	6
		3	06.06	2	0.00	0	0.00	0
		4	0.00	0	0.00	0	0.00	0
		5	0.00	0	0.00	0	0.00	0
Kruskal–Wallis test	*p* = 0.011		a		B		bc	

Different letters (a, b, c) indicate statistical difference (*p* ≤ 0.05); different letters (A, B, C) indicate statistical difference (*p* ≤ 0.01).

**Table 3 animals-12-01085-t003:** Broiler chicken production results, carcass yield, and the proportion of individual elements in chilled broiler carcasses (*n* = 18).

Item	C		E1		E2		*p*-Value
	$\bar{X}$	SD	$\bar{X}$	SD	$\bar{X}$	SD	
Body weight (kg)	3.40 ^C^	0.89	3.71 ^A^	0.20	3.68 ^B^	0.11	0.004
Body weight gain (kg)	3.36 ^c^	0.21	3.67 ^a^	0.20	3.64 ^b^	0.17	0.033
Carcass weight (kg)	2.47 ^b^	0.08	2.62 ^a^	0.14	2.62 ^a^	0.10	0.045
Slaughter yield (%)	72.5 ^A^	1.17	70.5 ^B^	0.82	71.4 ^AB^	0.89	0.010
Breast muscles (%)	31.2	1.39	30.93	1.84	31.07	2.2	0.969
Leg muscles (%)	19.61	1.93	18.63	3.33	19.73	1.66	0.613
Gizzard (%)	0.78	0.13	0.88	0.14	0.83	0.07	0.413
Heart (%)	0.58	0.07	0.59	0.04	0.59	0.11	0.501
Liver (%)	2.51	0.23	2.43	0.27	2.73	0.42	0.318
Abdominal fat (%)	0.74 ^a^	0.19	0.63 ^ab^	0.23	0.49 ^b^	0.16	0.039

^A, B, C^ statistically significant differences at *p* ≤ 0.01; ^a, b, c^ statistically significant differences at *p* ≤ 0.05.

**Table 4 animals-12-01085-t004:** Physicochemical properties and color of broiler breasts (*n* = 18).

Item	C		E1		E2		*p*-Value
	$\bar{X}$	SD	$\bar{X}$	SD	$\bar{X}$	SD	
pH24	5.56 ^B^	0.04	5.74 ^A^	0.03	5.71 ^AB^	0.08	0.024
Drip loss (%)	4.67	2.07	4.67	2.42	5.83	2.71	0.636
WHC (cm^2^/g)	2.82	0.69	3.25	0.53	3.50	0.70	0.210
L * brightness	60.33	1.93	61.26	4.53	61.70	1.47	0.719
a * redness	12.50	2.83	14.04	1.05	13.79	1.47	0.363
b * yellowness	10.27	3.99	9.64	0.88	11.05	1.07	0.615
ΔE C: E1, E2	0	-	1.91	-	2.03	-	
ΔE E1: E2	-	-	0	-	1.49	-	

Parameter L * (color brightness) can have values from 0 to 100. Parameters a * (redness) and b * (yellowness) are tri-chromaticity coordinates and can have positive and negative values: +a * corresponds to red, −a * to green, +b * to yellow, and −b * to blue. ΔE = absolute color difference. ^A, B^ statistically significant differences at *p* ≤ 0.01.

**Table 5 animals-12-01085-t005:** Selected enzyme activity and antioxidant potential for chosen substances in the breast muscles of broilers (*n* = 18).

Item	C		E1		E2		*p*-Value
	$\bar{X}$	SD	$\bar{X}$	SD	$\bar{X}$	SD	
AlaAP, nmol/mg protein/h	455.75	22.56	472.43	37.59	483.70	25.43	0.229
LeuAP, nmol/mg protein/h	175.68	28.52	177.36	10.50	198.00	31.86	0.229
ArgAP, nmol/mg protein/h	173.52 ^b^	16.46	185.31 ^ab^	3.72	191.23 ^a^	18.70	0.044
AcP, nmol/mg protein/h	1952.93	192.87	2082.36	233.48	1980.20	259.53	0.603
BGDR, nmol/mg protein/h	349.26	132.92	364.12	189.78	271.27	23.61	0.351
BGAL, nmol/mg protein/h	265.01 ^b^	19.35	292.70 ^ab^	29.51	295.71 ^a^	19.99	0.025
BGLU, nmol/mg protein/h	123.61 ^b^	11.15	150.43 ^a^	16.64	135.83 ^ab^	21.02	0.049
HEX, nmol/mg protein/h	612.72 ^B^	44.33	682.10 ^A^	35.72	697.83 ^A^	49.22	0.006
aGlu, nmol/mg protein/h	69.24	7.74	72.08	7.63	74.36	6.37	0.435
MAN, nmol/mg protein/h	154.30 ^c^	12.33	179.99 ^a^	23.51	172.43 ^b^	10.42	0.033
Vitamin C, mg/100 mL	2.86 ^a^	0.72	2.37 ^ab^	0.79	2.08 ^b^	0.53	0.045
DPPH, % of free radical scavenging activity	52.57	2.60	56.48	7.46	53.66	3.48	0.368
GSH, uMol/L	0.27	0.06	0.29	0.01	0.28	0.04	0.672

^A, B^ statistically significant differences at *p* ≤ 0.01; ^a, b, c^ statistically significant differences at *p* ≤ 0.05.

**Table 6 animals-12-01085-t006:** Selected enzyme activity and antioxidant potential for chosen substances in broiler blood serum (*n* = 18).

Item	C		E1		E2		*p*-Value
	$\bar{X}$	SD	$\bar{X}$	SD	$\bar{X}$	SD	
AlaAP, nmol/mg protein/h	20.26 ^B^	1.63	28.23 ^A^	4.45	24.41 ^AB^	4.16	0.006
LeuAP, nmol/mg protein/h	27.30 ^B^	3.19	31.46 ^A^	4.18	26.29 ^B^	2.83	0.031
ArgAP, nmol/mg protein/h	18.11 ^B^	1.90	21.90 ^A^	3.56	18.13 ^B^	1.88	0.018
AcP, nmol/mg protein/h	46.44	11.97	52.51	11.47	50.27	8.35	0.604
BGDR, nmol/mg protein/h	5.09	1.38	4.97	1.06	7.87	3.57	0.069
BGAL, nmol/mg protein/h	7.82 ^b^	1.43	7.89 ^ab^	1.45	10.69 ^a^	3.86	0.025
BGLU, nmol/mg protein/h	4.49 ^ab^	1.49	3.53 ^b^	1.49	7.02 ^a^	3.84	0.016
HEX, nmol/mg protein/h	13.46 ^b^	2.10	11.50 ^b^	2.05	17.05 ^a^	6.14	0.014
aGlu, nmol/mg protein/h	4.63 ^b^	0.88	5.36 ^ab^	1.26	6.83 ^a^	1.90	0.035
MAN, nmol/mg protein/h	6.36	2.10	5.65	1.23	8.20	3.42	0.184
Vit. C, mg/100 mL	8.09 ^b^	1.84	10.19 ^a^	1.07	8.85 ^ab^	1.19	0.018
DPPH, % of free radical scavenging activity	80.01 ^A^	2.87	74.43 ^B^	2.91	79.63 ^A^	3.56	0.011
GSH, uMol/L	1.11 ^b^	0.14	1.16 ^ab^	0.23	1.33 ^a^	0.16	0.040

^A, B^ statistically significant differences at *p* ≤ 0.01; ^a, b,^ statistically significant differences at *p* ≤ 0.05.

## Data Availability

All data generated or analyzed during the study are included in this published article. The datasets used and/or analyzed in the current study are available from the corresponding author on reasonable request.

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
