# Peer review of "Using Milk Thistle (Silybum marianum) Extract to Improve the Welfare, Growth Performance and Meat Quality of Broiler Chicken"

_animals, 2022, doi:10.3390/ani12091085_

Round 1
Reviewer 1 Report
In the Part2 (Materials and methods), the feed mixes composition and their nutritional value are missing.
In the Part 2 (Materials and methods) is necessary better explained and described of administration infusion of milk thistle. How many active substances were in the extract? How much active substance should the broiler take? How was it guaranteed that the broiler drank enough water daily to take the required amount of active substances? Was the drinking water continuously replenished? Has the active substances content of the extract been determined? Have you observed some losses in the thermo-labile content of silymarin? Whereas the temperature is already above 40 degrees Celsius can cause its degradation, while the infusion was prepared from dry seeds of milk thistle by boiling at a temperature of up to 90 degrees. Broiler weights were recorded at weekly intervals. In the discussion, I recommend adding the dynamics of body weights with focus on comparisons of the time before and after the 21st day of monitoring. Feed consumption was recorded at the end of the experiment or also at weekly intervals as the body weight of the chickens was monitored? Was the feed conversion ratio (FCR) calculated only at the end of the experiment? Or also continuously at weekly intervals?In the part (Results), in the section 3.2 is in the text the wrong table number is listed (the correct table number should be Table 4, not Table 3.
Author Response
Dear Reviewer 1
Authors are very grateful for valuable comments to improve our paper. We have implemented all the remarks indicated and believe the paper is now suitable for publication. To support visibility of the modifications made, we have introduced changes in the text by highlighting with yellow colour in the manuscript.
The following remarks have been addressed in the manuscript:
- In the Part2 (Materials and methods), the feed mixes composition and their nutritional value are missing.
- Line 92 and 98: Feed mixes and table has been added
- In the Part 2 (Materials and methods) is necessary better explained and described of administration infusion of milk thistle.
- Line 109: The administration infusion of milk thistle have explained and added.
- How many active substances were in the extract? How much active substance should the broiler take?
- Milk thistle for this experiment has taken from company that is why they did not give us specific information.
- How was it guaranteed that the broiler drank enough water daily to take the required amount of active substances?
- As it mentioned in Line 112, Birds drunk all amount dosage that have prepared for them daily, then clear water offered.
- Was the drinking water continuously replenished?
- Every day the determined dosage was prepared fresh for birds depend on selected concentration after that replenished by clear water.
- Has the active substances content of the extract been determined?
- No, the substances content have not determined because we just wanted to find the best level of milk thistle herbal to breeders for broiler chicken maybe in further study will do it as well.
- Have you observed some losses in the thermo-labile content of silymarin? Whereas the temperature is already above 40 degrees Celsius can cause its degradation, while the infusion was prepared from dry seeds of milk thistle by boiling at a temperature of up to 90 degrees.
- As we have read from articles that already done by authors. They mentioned that (Less than 100 °C will not cause losses of phytosterols and tocopherols and also resulted in changes in fatty acid composition [Marszałkiewicz, S., Siger, A., Gawrysiak-Witulska, M., Kmiecik, D., & Rudzińska, M. (2020). The effect of drying temperature of milk thistle seeds on quality and bioactive compounds in the lipid fraction. Journal of Food Science and Technology, 57(11), 4003-4013.]).
- Broiler weights were recorded at weekly intervals.
- No, body weight was recorded depend of program feeding as it has already added in Line 94. (At the end of each period the birds were weighed).
- In the discussion, I recommend adding the dynamics of body weights with focus on comparisons of the time before and after the 21st day of monitoring.
- Suggested changes have done in Line 351.
- Feed consumption was recorded at the end of the experiment or also at weekly intervals as the body weight of the chickens was monitored? Was the feed conversion ratio (FCR) calculated only at the end of the experiment? Or also continuously at weekly intervals?
- Same as body weight as already mentioned in Line 94. The feed consumption has calculated four times depend of the changing stage of feed (Starter, Grower I, Grower II and Finisher).
- In the part (Results), in the section 3.2 is in the text the wrong table number is listed (the correct table number should be Table 4, not Table 3.
- The mistake corrected already.
Regards,
Reviewer 2 Report
Dear Corresponding Author,
Please, respond to all reviewer's queries and suggestions listed line-by-line below.
L6-18 - please, do not use private mail addresses, such as gmail.com; only institutional are permitted.
Simple summary
L22 - selected or chosen concentrations instead of different
L25 - "some of the analyzed parameters." - it is too general please, be more precise.
Abstract
Comment 1 - there is a lack of information about the experimental birds used in the study, i.e., sex, hybrid, age, how many replications was used, and the number of birds in each replication.
Introduction
L56 - "liver. [9,10].| please, remove dot.
Material and methods
L73 - the Authors mentioned that 102 male chicks were used in the study, however, in line 34 there is highlighted n=103. Please, explain.
L76 - the composition of the diet should be presented in the separate Table. Currently, no diet characteristic has been applied, including the composition, nutritive value, physical form, etc.
L76 - "starter, grower, and finisher." please, add information "from... to..."
L77 - "standard conditions, with a light cycle", please be precise. Add information about the environmental conditions briefly and add references.
L78 "slaughtered" - please, explain how the process was done? At the very beginning of the section please add information about the Ethical Committee approval (please, move paragraphs L80-83 to L72).
Comment 2 - there is no information on where the birds were kept, e.g., in cages? in boxes? in artificial conditions? or maybe in the chicken house? Additionally, according to which nutritional requirements were birds fed? Please, add this information.
Comment 3 - how many nipples were used in the replication? how many birds had access to one nipple...
Table 1. From the Reviewer's point of view Table, 1 is not necessary and should be removed. The presentation of the dosage should appear in the text.
Comment 4 - According to the "Milk thistle dosage for chickens" headings, it is crucial to add characterization of the experimental factor, particularly when the Authors mention the bioactive compounds present in the Milk thistle extract.
Comment 5 - in the material and method section, there is no information/explanation why the Authors used exactly these doses.
Comment 6 - There is no information about the experimental unit used in terms of growth performance results; it means that the individual birds were defined as an exp. unit? What about the FCR, from the reviewer's point of view it is not good to use average feed intake to calculate the FCR. The Authors used only 2 replicates which is unacceptable in the case of the growth performance calculations. Please, explain; however, the reviewer wants to suggest excluding the growth performance results.
Comment 7 - please, explain why the Authors used the "NIR method" to determine the chemical composition of meat, not AOAC methods?
L163 - how the blood was collected? post-mortem, before slaughter, wing vein?
Comment 8 - in the whole Material and Methods section there is no information about the experimental units defined to determine each parameter. The 'n' number should also appear in the text.
L200 - Tukey post-hoc test was used
Comment 9 - there is a need to add information about the homogeneity of variance test used in the study. The Authors did not inform which post-hoc test was chosen when Kruscal-Wallis test was used. Please, explain why the Kendall tau-b correlation was used, not the spearman correlation?
Results
L208 - please, add the exact p-value
Figure 1. From the reviewer's point of view Figure poorly present the data. Firstly, there is no explanation of what the numbers inside the columns present. Probably it is a number of chicks, however, there is no explanation why 99 birds, not 102, as mentioned in line 73. The reviewer does not understand why the significance appears; each column has 100%, thus the significance is probably related to gait 0? But there is no information about that. What about the significance related to gait 1,2 and 3? The figure should be remodeled.
Figure 2 - the same as above.
Table 2 - is not necessary and should be deleted. More information is presented in L228-229. Additionally, why n=99?
L229 - the exact p-value should be given.
Comment 10 - as the reviewer mentioned above, the section about the growth performance should be removed, due to no possibility to obtain the individual feed intake and usage of two replication for each treatment.
Comment 11 - the whole results section should be revised and only the exact p-value can be used.
Table 4 - please, add a column with the p-value. Furthermore, there is no information about the absolute mass or in relation to BW. This must be added n the text and in the Table.
Table 5, 6, 7 - as above
Discussion
Comment 12 - due to the recommendation of the reviewer, please remove the paragraph about the growth performance from the discussion section.
L330-333 - the discussion about the slaughter yield results should be expanded, please propose the mode of action.
Conclusions
As above, the Authors cannot include the FCR, thus only body weight gain can be concluded.
Author Response
Dear Reviewer 2
Authors are very grateful for valuable comments to improve our paper. We have implemented all the remarks indicated and believe the paper is now suitable for publication. To support visibility of the modifications made, we have introduced changes in the text by highlighting with green colour in the manuscript.
The following remarks have been addressed in the manuscript:
- L6-18 - please, do not use private mail addresses, such as gmail.com; only institutional are permitted.
- Mail addresses have changed in Line 8, 9.
Simple summary
- L22 - selected or chosen concentrations instead of different
- Suggested changes have already done in Line 23.
- L25 - "some of the analyzed parameters." - it is too general please, be more precise.
- Suggested changes have already done in Line 27.
Abstract
Comment 1 - there is a lack of information about the experimental birds used in the study, i.e., sex, hybrid, age, how many replications was used, and the number of birds in each replication.
- That information’s have added in abstract in Line 34, 35.
Introduction
L56 - "liver. [9,10].| please, remove dot.
- The dot removed already in Line 61.
Material and methods
L73 - the Authors mentioned that 102 male chicks were used in the study, however, in line 34 there is highlighted n=103. Please, explain.
- The mistake has corrected in Line 34.
L76 - the composition of the diet should be presented in the separate Table. Currently, no diet characteristic has been applied, including the composition, nutritive value, physical form, etc.
- The composition of the diet added in table 1, from Line 98.
L76 - "starter, grower, and finisher." please, add information "from... to..."
- Added already in Line 98.
L77 - "standard conditions, with a light cycle", please be precise. Add information about the environmental conditions briefly and add references.
- Line 85, Suggested information added.
L78 "slaughtered" - please, explain how the process was done? At the very beginning of the section please add information about the Ethical Committee approval (please, move paragraphs L80-83 to L72).
- The slaughter process added and explained in Line 87.
Comment 2 - there is no information on where the birds were kept, e.g., in cages? in boxes? in artificial conditions? or maybe in the chicken house? Additionally, according to which nutritional requirements were birds fed? Please, add this information.
- The information has added already in the text (Line 84) and the nutrition requirement was according to ROSS 308 guide which can be seen in Table 1.
Comment 3 - how many nipples were used in the replication? how many birds had access to one nipple...
- Bell Drinkers were used in this study. The size of bell drinkers were 38 centimeter of diameter have used for 17 birds.
Table 1. From the Reviewer's point of view Table, 1 is not necessary and should be removed. The presentation of the dosage should appear in the text.
- Suggested changes have done and the information has written in Line 109.
Comment 4 - According to the "Milk thistle dosage for chickens" headings, it is crucial to add characterization of the experimental factor, particularly when the Authors mention the bioactive compounds present in the Milk thistle extract.
- Milk thistle for this experiment has taken from company that is why they did not give us specific information. So depend of previous studies that they have already showed in their papers and we presented in introduction part as well.
Comment 5 - in the material and method section, there is no information/explanation why the Authors used exactly these doses.
- Depend on previous study we decide to choose that doses.
Comment 6 - There is no information about the experimental unit used in terms of growth performance results; it means that the individual birds were defined as an exp. unit? What about the FCR, from the reviewer's point of view it is not good to use average feed intake to calculate the FCR. The Authors used only 2 replicates which is unacceptable in the case of the growth performance calculations. Please, explain; however, the reviewer wants to suggest excluding the growth performance results.
- As we mentioned in material and methods part that we have replicates, so for body weight, feed intake and FCR which recorded depend of feeding program: starter, days 1–10; grower I, days 11–24; grower II, days 25–35; and finisher, days 36–42. The growth performance parameters have been recorded at the last day for each period. The authors only used two repetitions because the experiment was pilot and the ethics committee always recommends the lowest number of birds.
Comment 7 - please, explain why the Authors used the "NIR method" to determine the chemical composition of meat, not AOAC methods?
- Normally we determine chemical composition by NIR method because we have this machine in our faculty it will be cheaper for us. Moreover, we have good calibration for that.
L163 - how the blood was collected? post-mortem, before slaughter, wing vein?
- Blood was collected after slaughter (post-mortem) Line 192.
Comment 8 - in the whole Material and Methods section there is no information about the experimental units defined to determine each parameter. The 'n' number should also appear in the text.
- Suggested information has added and they can be seen in Line 120, 151, 166 and 192.
L200 - Tukey post-hoc test was used
- Corrected in Line 233
Comment 9 - there is a need to add information about the homogeneity of variance test used in the study. The Authors did not inform which post-hoc test was chosen when Kruscal-Wallis test was used. Please, explain why the Kendall tau-b correlation was used, not the spearman correlation?
- Kendall's tau-b correlations were used because the study variables were measured on an ordinal scale and there were many tied ranks (more than 1/3 of the results).
Results
L208 - please, add the exact p-value
- All exact p-value have added in the text and highlighted by green colour.
Figure 1. From the reviewer's point of view Figure poorly present the data. Firstly, there is no explanation of what the numbers inside the columns present. Probably it is a number of chicks, however, there is no explanation why 99 birds, not 102, as mentioned in line 73. The reviewer does not understand why the significance appears; each column has 100%, thus the significance is probably related to gait 0? But there is no information about that. What about the significance related to gait 1,2 and 3? The figure should be remodeled.
- Actually the total number was 102 but three birds dead that is why it remains 99.
- Figure has been changed in Line 251.
Figure 2 - the same as above
- Figure has been changed in Line 251.
Table 2 - is not necessary and should be deleted. More information is presented in L228-229. Additionally, why n=99?
- Table 2 deleted, the number is 99 because three chickens dead.
L229 - the exact p-value should be given.
- P-value have added.
Comment 10 - as the reviewer mentioned above, the section about the growth performance should be removed, due to no possibility to obtain the individual feed intake and usage of two replication for each treatment.
- As we mentioned in material and methods part that we have replicates, so for body weight, feed intake and FCR which recorded depend of feeding program: starter, days 1–10; grower I, days 11–24; grower II, days 25–35; and finisher, days 36–42. The growth performance parameters have been recorded at the last day for each period.
Comment 11 - the whole results section should be revised and only the exact p-value can be used.
- Done and all exact p-value added.
Table 4 - please, add a column with the p-value. Furthermore, there is no information about the absolute mass or in relation to BW. This must be added n the text and in the Table.
- Added already Line 278.
Table 5, 6, 7 - as above
- Done in Line 289, 303 and 316.
Discussion
Comment 12 - due to the recommendation of the reviewer, please remove the paragraph about the growth performance from the discussion section.
- Deleted already.
L330-333 - the discussion about the slaughter yield results should be expanded, please propose the mode of action.
- Suggested changes have done in Line 366-370.
Conclusions
As above, the Authors cannot include the FCR, thus only body weight gain can be concluded.
- Suggested changes have done, FCR already deleted.
Regards,

Reviewer 3 Report
Dear Authors,
Please see my comments below:
- The title does not present the study well. The title must be a representative of all data.
- Both summary and abstract are well written and easy to follow. However, it would be better to add more results or explain the method a bit more in the abstract.
- Line 76. Some information are required for the feed (CP, energy and etc) for each one, starter , grower and finisher.
- Light provided 24h/day? RH?
- What was the bedding?
- How could you explain the diet impacted food pad score not other factors? Changing saw dust or available moisture could have a significant impact of the results regardless of the diet.
- There is not other data to confirm the scoring results. I would recommend you to revise the title of manuscript and remove "Welfare". Unless you could provide more data to support the current results.
- Figure 2. It requires to explain what are 0-2 (colored). The figure must be easy to read even alone.
- I have no comments on the discussion, however, the conclusion section is missing some important results like FP score.
Regards,
Author Response
Dear Reviewer 3
Authors are very grateful for valuable comments to improve our paper. We have implemented all the remarks indicated and believe the paper is now suitable for publication. To support visibility of the modifications made, we have introduced changes in the text by highlighting with pink colour in the manuscript.
The following remarks have been addressed in the manuscript:
- The title does not present the study well. The title must be a representative of all data.
- Suggested changes have done in title (Line 3)
- Both summary and abstract are well written and easy to follow. However, it would be better to add more results or explain the method a bit more in the abstract.
- Same comments from other reviewer, some information have already added and highlighted but because of word limitation we could not explain more about them in abstract. (Line 41)
- Line 76. Some information are required for the feed (CP, energy and etc) for each one, starter , grower and finisher.
- Suggested information have already added about feed for all stages in Line, 98.
- Light provided 24h/day? RH?
- The information has added in Line, 85.
- What was the bedding?
- The information has added in Line 84-85.
- How could you explain the diet impacted food pad score not other factors? Changing saw dust or available moisture could have a significant impact of the results regardless of the diet.
- Suggested idea already has added in Line, 333.
- There is not other data to confirm the scoring results. I would recommend you to revise the title of manuscript and remove "Welfare". Unless you could provide more data to support the current results.
- We are totally agree with your recommendation but for this study we have done three parameters that relate with welfare maybe for future study will do more about it.
- Figure 2. It requires to explain what are 0-2 (colored). The figure must be easy to read even alone.
- Figures have revised and arranged again (Line 251).
- I have no comments on the discussion; however, the conclusion section is missing some important results like FP score.
- Result about FPD have added in conclusion (Line 422-424).
Regards,

Round 2
Reviewer 2 Report
Dear Corresponding Author,
Thank you for considering the previous comments.
In this stage, only a few minor changes should be made.
Table 1 - the AME - apparent metabolizable energy instead of "EM"; Crude protein instead of protein; Crude fat instead of Fat; Calculated chemical composition instead of "Nutritional composition"
Table 2 - The presentation of FPD and gait score results are still quite enigmatic... because of the p-value. Letters A, AB, and C are related to which result? Score 0? or maybe 2? They indicate the difference between what? It should be clearly stated.
I suggest using the same result presentation as Kheravii et al. 2017.
Dietary sugarcane bagasse and coarse particle size of corn are beneficial to performance, and gizzard development in broilers fed normal and high sodium diets. Poultry Sciences
Comment 1 - each table in the footer should have a number of replicates "n=?"
Author Response
Dear Reviewer 2
Authors are very grateful for valuable comments to improve our paper. We have implemented all the remarks indicated and believe the paper is now suitable for publication. To support visibility of the modifications made, we have introduced changes in the text by highlighting with yellow colour in the manuscript.
The following remarks have been addressed in the manuscript:
- Table 1 - the AME - apparent metabolizable energy instead of "EM"; Crude protein instead of protein; Crude fat instead of Fat; Calculated chemical composition instead of "Nutritional composition"
Suggested changes have done in Table 1.
- Table 2 - The presentation of FPD and gait score results are still quite enigmatic... because of the p-value. Letters A, AB, and C are related to which result? Score 0? or maybe 2? They indicate the difference between what? It should be clearly stated
In the opinion of the authors, the way the results of FPD and GS are presented in the present form best represents the results obtained. We modelled the presentation of the results on the paper: Animal-based welfare indicators of 4 slow-growing broiler genotypes for the approval in an animal welfare label program (Louton et al., 2019; Poultry Science). The Kruskal-Wallis test conducted verified the H0 assuming that each of the diets used gives the same results as to the distribution of scores in FPD and GS evaluation, hence, the use of A B C labels does not refer to a specific score value of 0, 1 or 2, but to the distribution of scores given in the groups studied. This test is based on the ranks of the observations. For example, if all samples are from the same population, we expect the average ranks in each group to be similar.
In the presentation of results proposed by reviewer 2 with reference to the work of the authors Kheravia et al. 2017, a classical table for the presentation of results for the General Linear Model was used, which is not adequate with the statistical method we used. In our table, the recipient of the results can additionally easily verify the number of foot evaluated for a specific grade and what percentage of the total they represent in the study group.
- Comment 1 - each table in the footer should have a number of replicates "n=?"
Suggested changes have done in Table 2-6.
Regards,

Reviewer 3 Report
Dear authors,
Thanks for the revised version. However, the following statement requires citation:
- Line 84. "were maintained according to ROSS 308 standard management guide"
Regards,
Author Response
Dear Reviewer 3
Authors are very grateful for valuable comments to improve our paper. We have implemented all the remarks indicated and believe the paper is now suitable for publication. To support visibility of the modifications made, we have introduced changes in the text by highlighting with yellow colour in the manuscript.
The following remarks have been addressed in the manuscript:
- Thanks for the revised version. However, the following statement requires citation: Line 84. "were maintained according to ROSS 308 standard management guide"
Suggested changes have done in Line 84.
Regards,
